# Deconvoluting heme biosynthesis to target blood-stage malaria parasites

**Paul A Sigala[1,2], Jan R Crowley[3], Jeffrey P Henderson[1,2,3], Daniel E Goldberg[1,2]\***

[1]Department of Molecular Microbiology, Washington University School of Medicine, St Louis, United States; [2]Department of Medicine Division of Infectious Diseases, Washington University School of Medicine, St Louis, United States; [3]Center for Women's Infectious Disease Research, Washington University School of Medicine, St. Louis, United States

**Abstract** Heme metabolism is central to blood-stage infection by the malaria parasite *Plasmodium falciparum*. Parasites retain a heme biosynthesis pathway but do not require its activity during infection of heme-rich erythrocytes, where they can scavenge host heme to meet metabolic needs. Nevertheless, heme biosynthesis in parasite-infected erythrocytes can be potently stimulated by exogenous 5-aminolevulinic acid (ALA), resulting in accumulation of the phototoxic intermediate protoporphyrin IX (PPIX). Here we use photodynamic imaging, mass spectrometry, parasite gene disruption, and chemical probes to reveal that vestigial host enzymes in the cytoplasm of *Plasmodium*-infected erythrocytes contribute to ALA-stimulated heme biosynthesis and that ALA uptake depends on parasite-established permeability pathways. We show that PPIX accumulation in infected erythrocytes can be harnessed for antimalarial chemotherapy using luminol-based chemiluminescence and combinatorial stimulation by low-dose artemisinin to photoactivate PPIX to produce cytotoxic reactive oxygen. This photodynamic strategy has the advantage of exploiting host enzymes refractory to resistance-conferring mutations.

## Introduction

Malaria is an ancient and deadly scourge of humanity, with hundreds of millions of people infected by *Plasmodium* malaria parasites and more than 600,000 deaths each year (*WHO, 2014*). Substantial progress has been made in reducing the global malaria burden, due in part to the success of artemisinin-based combination drug therapies (ACTs). Recent identification of artemisinin-tolerant parasites in southeast Asia, however, has raised concerns that the broad potency of ACTs against all parasite strains may be waning, which could lead to a resurgence in malaria deaths (*Dondorp et al., 2009*; *Ariey et al., 2014*). These concerns motivate continued efforts to deepen understanding of basic parasite biology in order to identify new drug targets and facilitate development of novel therapies.

Heme is a ubiquitous biological cofactor required by nearly all organisms to carry out diverse redox biochemistry (*Ponka, 1999*). Heme metabolism is a dominant feature during *Plasmodium* infection of erythrocytes, the most heme-rich cell in the human body and the stage of parasite development that causes all clinical symptoms of malaria. Parasites sequester and biomineralize the copious heme released during large-scale hemoglobin digestion in their acidic food vacuole (*van Dooren et al., 2012*; *Sigala and Goldberg, 2014*); they also require heme as a metabolic cofactor for cytochrome-mediated electron transfer within mitochondria (*Painter et al., 2007*; *van Dooren et al., 2012*; *Sigala and Goldberg, 2014*).

Sequencing of the *Plasmodium falciparum* genome over a decade ago and subsequent studies have revealed that parasites encode and express all of the conserved enzymes for a complete heme biosynthesis pathway (*Figure 1A*), but the role and properties of this pathway have been the subject of considerable confusion and uncertainty (*Gardner et al., 2002*; *van Dooren et al., 2012*; *Sigala and Goldberg, 2014*). This pathway was originally proposed to be essential for blood-stage parasite

**\*For correspondence:**
goldberg@wusm.wustl.edu

**Competing interests:**
See page 17

**eLife digest** Malaria is a devastating infectious disease that is caused by single-celled parasites called *Plasmodium* that can live inside red blood cells. Several important proteins from these parasites require a small molecule called heme in order to work. The parasites have enzymes that make heme via a series of intermediate steps. However, it remains unclear exactly how important this 'pathway' of enzymes is for the parasite, and whether this pathway could be targeted by drugs to treat malaria.

Now Sigala et al. have used a range of genetic and biochemical approaches to better understand the production of heme molecules in *Plasmodium*-infected red blood cells. First, several parasite genes that encode the enzymes used to make heme molecules were deleted. Unexpectedly, these gene deletions did not affect the ability of the infected blood cells to make heme. This result suggested that the parasites do not use their own pathway to produce heme while they are growing in the bloodstream. Sigala et al. then showed that human enzymes involved in making heme, most of which are also found within the infected red blood cells, are still active. These human enzymes provide a parallel pathway that can link up with the final parasite enzyme to generate heme.

Further experiments revealed that the activity of the human enzymes could be strongly stimulated by providing the pathway with one of the building blocks used to make heme. This stimulation led to the build-up of an intermediate molecule called PPIX. This intermediate molecule can kill cells when it is exposed to light—a property that is called 'phototoxicity'. Sigala et al. showed that treating infected red blood cells with a new combination of non-toxic chemicals that emit light can activate PPIX in the bloodstream and can selectively kill the malaria parasites while leaving uninfected cells intact. These findings suggest a new treatment that could be effective against blood-stage malaria. Furthermore, the parasite will be unable to easily mutate to avoid the effects of this treatment because it relies on human proteins that are already made. Future work is now needed to optimize the dosage and the combination of drugs that could provide such a treatment.

development and thus a potential drug target (*Surolia and Padmanaban, 1992*), as host heme was thought to be inaccessible for parasite utilization in mitochondria. Recent studies, however, have clarified that de novo heme synthesis is not required by intraerythrocytic parasites and therefore is unlikely to be a viable target for therapeutic inhibition (*Nagaraj et al., 2013*; *Ke et al., 2014*). The parasite-encoded ferrochelatase (FC) can be knocked out to ablate heme biosynthesis but parasite growth is unaffected, suggesting that parasites can scavenge host heme to satisfy metabolic requirements during blood-stage infection.

Here we use chemical and physical probes to decipher the role of upstream enzymes in heme biosynthesis by parasite-infected erythrocytes. Contrary to simple predictions, genetic disruption of the parasite porphobilinogen deaminase (PBGD) and coproporphyrinogen III oxidase (CPO) had no effect on the ability of *Plasmodium*-infected erythrocytes to convert exogenous aminolevulinic acid (ALA) into heme, as monitored by mass spectrometry and photodynamic imaging of porphyrin intermediates. Biochemical fractionation revealed that vestigial host enzymes remaining in the erythrocyte cytoplasm contribute to ALA-stimulated heme biosynthesis, explaining why disruption of parasite enzymes had no effect on biosynthetic flux. We used small-molecule probes to show that ALA uptake by erythrocytes, which are normally impermeable to ALA, requires the parasite nutrient-acquisition pathways established after invasion. Finally, we show that latent host enzyme activity can be exploited for antimalarial photodynamic therapy (PDT) using (1) ALA to stimulate production of the phototoxic intermediate protoporphyrin IX (PPIX), (2) luminol-based chemiluminescence to photoactivate PPIX cytotoxicity within infected erythrocytes, and (3) luminol stimulation by combinatorial low-dose artemisinin.

## Results

### Exogenous ALA stimulates heme biosynthesis and photosensitizes parasites

Production of 5-ALA by ALA synthase (ALAS) is the regulated step in heme biosynthesis. Exogenous ALA bypasses this step and stimulates biosynthetic flux, leading to accumulation of the final

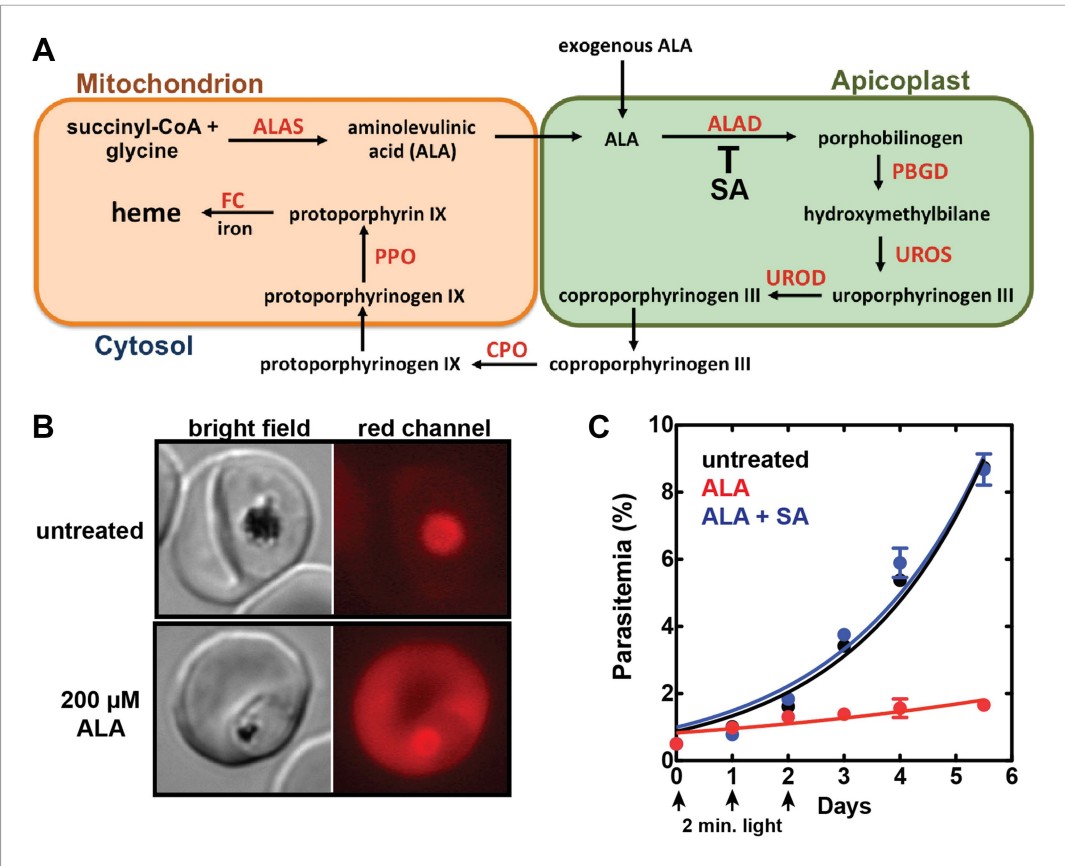

**Figure 1**. Exogenous ALA stimulates protoporphyrin IX (PPIX) biosynthesis in Plasmodium-infected erythrocytes. (**A**) Schematic depiction of the heme biosynthesis pathway in *Plasmodium falciparum* parasites. Enzymes abbreviations are in red and pathway substrates and intermediates are in black: ALAS (aminolevulinic acid synthase), ALAD (aminolevulinic acid dehydratase), PBGD (porphobilinogen deaminase), UROS (uroporphyrinogen synthase), UROD (uroporphyrinogen decarboxylase), CPO (coproporphyrinogen III oxidase), PPO (PPIX oxidase), FC (ferrochelatase). For simplicity, all organelles are depicted with single membranes. Succinylacetone (SA) inhibits ALAD. (**B**) Bright field and fluorescence microscopy images of untreated and 200 μM aminolevulinic acid (ALA)-treated parasites. Fluorescence images were acquired with a Zeiss filter set 43 HE (excitation 537–562 nm, emission 570–640 nm). (**C**) Growth of asynchronous 3D7 parasites in the presence or absence of 200 μM ALA and 50 μM SA, with 2-min exposures to white light on an overhead projector on days 0–2. Parasitemia (percentage of total erythrocytes infected with parasites) as a function of time was fit with an exponential growth equation.

The following figure supplements are available for figure 1:

**Figure supplement 1**. Fluorescence excitation and emission spectrum of PPIX in aqueous buffer.

**Figure supplement 2**. Transmission electron microscopy images of untreated and 500 μM ALA-treated *P. falciparum*-infected erythrocytes after light exposure on an overhead projector light box.

**Figure supplement 3**. ALA dose dependence of photosensitivity by blood-stage Plasmodium parasites.

**Figure supplement 4**. Giemsa-stained blood smear of *P. falciparum* culture after 3 days of treatment with 200 μM ALA and 2-min daily light exposure on an overhead projector light box.

intermediate, PPIX, since conversion of PPIX to heme by FC becomes rate limiting (*Kennedy et al., 1990*). PPIX is fluorescent (*Figure 1—figure supplement 1*) and its cellular accumulation can be directly visualized by fluorescence microscopy using a phenomenon called 'photodynamic imaging' that has been exploited for visualizing cancerous tumor boundaries during surgical resection (*Celli et al., 2010*).

Prior work indicated that exogenous ALA stimulates PPIX production in *P. falciparum* parasites (*Smith and Kain, 2004*; *Ke et al., 2014*). We therefore posited that ALA treatment could serve as a probe of heme biosynthesis activity in *Plasmodium*-infected erythrocytes by enabling direct visualization of PPIX production and the cellular consequences of light activation. Untreated parasites imaged on an epifluorescence microscope display only background auto-fluorescence from hemozoin crystals in the parasite digestive vacuole (*Figure 1B*). Parasites grown in 200 µM ALA, however, display bright red fluorescence distributed throughout the infected erythrocyte, as expected for accumulation of PPIX (*Figure 1B*).

PPIX is also known to generate cytotoxic reactive oxygen species when photo-illuminated, owing to formation of an excited triplet state that can undergo collisional energy transfer with ground state triplet oxygen to produce highly reactive singlet oxygen (*Celli et al., 2010*). The ability to kill ALA-treated cells by light activation of accumulating PPIX has been exploited to selectively target cancerous tumors via a strategy known as 'photodynamic therapy' (*Kennedy et al., 1990*; *Celli et al., 2010*).

The cytotoxic effects of light-activating PPIX were readily apparent by monitoring the motion of hemozoin crystals that dynamically tumble within the digestive vacuole of individual parasites. Although the origin and physiological significance of this motion remain unknown, hemozoin dynamics serve as an internal biomarker of food vacuole integrity and parasite viability (*Sigala and Goldberg, 2014*). Hemozoin motion in untreated parasites was unaffected by exposure to light that overlaps the excitation wavelength of PPIX (*Videos 1, 2*). In contrast, the hemozoin dynamics in ALA-treated parasites were rapidly ablated by light (*Videos 3, 4*). Ultra-structural analysis by electron microscopy revealed a loss of electron density within the digestive vacuole of ALA-treated and illuminated parasites (*Figure 1—figure supplement 2*), suggesting disruption of the food vacuole membrane and outward diffusion of the vacuolar protein contents. These changes are consistent with a photodynamic mechanism of PPIX-mediated generation of reactive oxygen species that cause pleiotropic cytotoxic damage, including loss of lipid bilayer integrity.

To test the effects of these changes on bulk parasite culture, we monitored the growth of untreated vs ALA-treated parasites over several days with 2-min daily exposures to broad-wavelength white light on an overhead projector light box. Whereas untreated parasite cultures grew normally in the presence of light, the growth of ALA-treated cultures was strongly attenuated by light (*Figure 1C* and *Figure 1—figure supplement 3*), consistent with a prior report (*Smith and Kain, 2004*). Microscopic examination by blood smear revealed that ALA-treated cultures predominantly contained karyolytic and pyknotic parasites (*Figure 1—figure supplement 4*), indicative of widespread cell death. The photosensitivity of parasite growth in ALA was fully rescued by 50 µM succinylacetone (SA), an ALA dehydratase (ALAD) inhibitor shown in previous work to substantially reduce PPIX biosynthesis from ALA in parasite-infected erythrocytes (*Nagaraj et al., 2013*; *Ke et al., 2014*). This chemical rescue confirmed that parasite photosensitivity in ALA requires biosynthetic conversion of ALA to PPIX.

## Stable disruption of parasite enzymes or the apicoplast organelle does not affect heme biosynthesis from ALA

The constituent enzymes of the parasite's heme biosynthesis pathway are distributed between three sub-cellular compartments: the mitochondrion, the cytoplasm, and the chloroplast-like but non-photosynthetic apicoplast organelle (*Figure 1A*) (*van Dooren et al., 2012*). Prior work indicated that the parasite FC gene could be knocked out, resulting in a complete ablation of de novo heme synthesis but with no effect on parasite growth (*Nagaraj et al., 2013*; *Ke et al., 2014*). This observation provided strong evidence that parasites do not require heme biosynthesis for growth in erythrocytes, where they can presumably scavenge abundant host heme to meet metabolic needs. We therefore reasoned that upstream enzymes in the parasite-encoded heme biosynthesis pathway should also be amenable to genetic disruption and that such ablations would block production of PPIX from exogenous ALA (*Figure 1A*) and thus prevent parasite photosensitivity.

**Video 1.** Hemozoin dynamics in the digestive vacuole of untreated parasites prior to light exposure.

**Video 2.** Hemozoin dynamics in the digestive vacuole of untreated parasites after light exposure.

**Video 3.** Hemozoin dynamics in the digestive vacuole of 200 μM ALA-treated parasites prior to light exposure. DOI: 10.7554/eLife.09143.010

**Video 4.** Hemozoin dynamics in the digestive vacuole of 200 μM ALA-treated parasites after light exposure. DOI: 10.7554/eLife.09143.011

We successfully disrupted the parasite genes encoding the apicoplast-targed PBGD (*Figure 2—figure supplement 1*) and the cytosolic CPO (*Figure 2—figure supplement 2*) using single-crossover homologous recombination to truncate the open reading frame for each gene. Southern blot and polymerase chain reaction (PCR) analysis confirmed correct integration and gene disruption in clonal parasite lines (*Figure 2—figure supplements 3, 4*). Contrary to simple predictions, however, these genetic disruptions had no effect on the ability of parasite-infected erythrocytes to incorporate $^{13}$C-labelled ALA into heme, PPIX, or copropor-phyrinogen III (CPP) (*Figure 2A*), as monitored by a previously developed LC-MS/MS assay (*Ke et al., 2014*), and clonal growth of both parasite lines remained fully photosensitive in ALA (*Figure 2B,C*).

To further probe the functional contributions to heme biosynthesis by the parasite apicoplast and its constituent enzymes, we stably disrupted this organelle by treating parasites with doxycycline and isopentenyl pyrophosphate (IPP). Doxycycline inhibits the prokaryotic translation machinery of the apicoplast, which blocks replication of the small apicoplast genome, prevents organelle segregation, and results in unviable apicoplast-deficient parasite progeny (*Dahl et al., 2006*). The lethal effects of doxycycline, however, can be chemically rescued by IPP, which enables parasites to make essential isoprenoids despite apicoplast disruption. This rescue leads to a stable metabolic state in which parasites lack the intact organelle such that nuclear-encoded proteins ordinarily targeted to the apicoplast become stranded in small vesicles (*Yeh and DeRisi, 2011*). We confirmed apicoplast loss in doxycycline and IPP-treated parasites by using microscopy to verify disrupted apicoplast targeting of the nuclear-encoded ALAD enzyme (*Figure 2D* and *Figure 2—figure supplement 5*), PCR to confirm loss of the small apicoplast genome (*Figure 2—figure supplement 6*), and western blot (WB) to visualize disruption of post-translational processing of ALAD (*Figure 2—figure supplement 7*). Despite apicoplast disruption, these parasites retained their capacity for heme biosynthesis (*Figure 2A*) and remained fully photosensitive in ALA (*Figure 2E*). Together with the gene disruption results above, these observations strongly suggested that parasite-infected erythrocytes have a parallel biosynthetic pathway that bypasses functional disruption of the parasite enzymes targeted to the apicoplast and cytoplasm.

## Erythrocytes retain vestigial heme biosynthesis enzymes with latent activity stimulated by ALA

The heme biosynthesis pathway in human cells is distributed between mitochondria and the cytoplasm, with four cytosolic enzymes that correspond to the four apicoplast-targeted enzymes in *Plasmodium* parasites (*Ponka, 1997*, *1999*; *van Dooren et al., 2012*). During human erythropoiesis, precursor reticulocytes carry out prolific heme biosynthesis, but this activity is absent in mature erythrocytes due to loss of mitochondria and their constituent heme biosynthesis enzymes, including ALAS and FC (*Ponka, 1997*). Proteomic studies have confirmed that mature erythrocytes retain the cytosolic enzymes (ALAD, PBGD, uroporphyrinogen synthase [UROS], and uroporphyri-nogen decarboxylase [UROD]) (*Pasini et al., 2006*; *D'alessandro et al., 2010*), but this vestigial pathway is ordinarily quiescent due to the lack of ALA synthesis or uptake in erythrocytes. We hypothesized that exogenous ALA taken up by parasite-infected erythrocytes might stimulate the latent activity of these cytosolic human enzymes, resulting in biosynthetic flux through this truncated host pathway and production of downstream tetrapyrrole intermediates that could be taken up by the parasite via hemoglobin import or other mechanisms and converted into heme within the parasite mitochondrion.

The cytosolic human enzymes remaining in the mature erythrocyte would be expected to produce CPP from ALA. Our observation that disruption of the parasite CPO had no effect on conversion of ALA into heme by intraerythrocytic parasites suggests that a soluble fraction of human CPO, which is

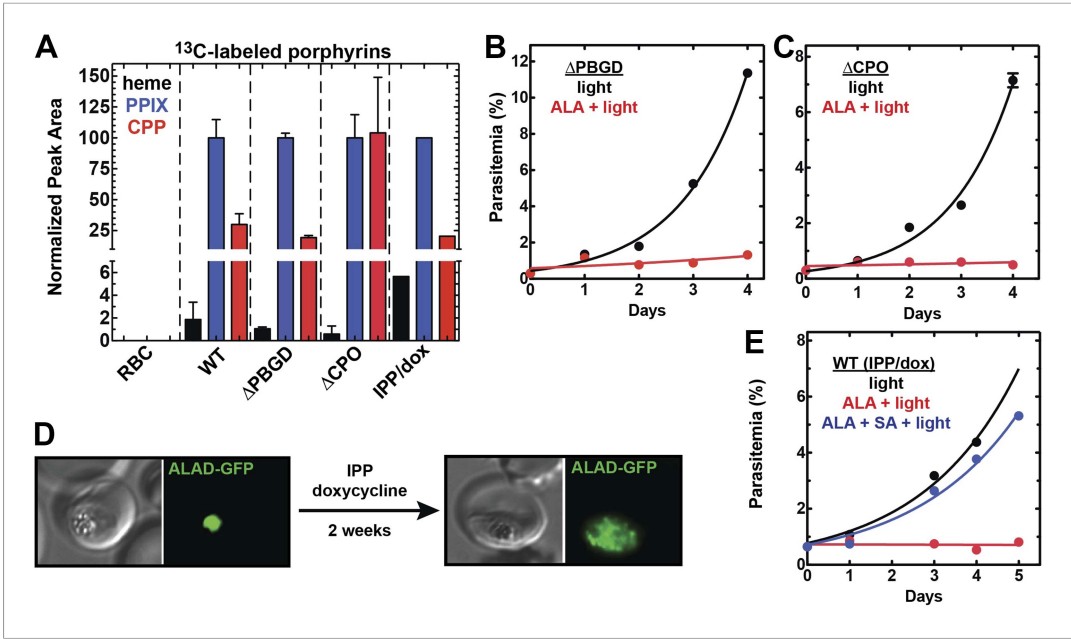

**Figure 2**. Heme biosynthesis in infected erythrocytes persists despite disruption of parasite enzymes or the apicoplast. (**A**) Liquid chromatography-tandem mass spectrometry (LC-MS/MS) detection of $^{13}$C-labelled heme, PPIX, and coproporphyrinogen III (CPP) in parasites grown in 200 μM 5-[$^{13}C_4$]-ALA. Parasites were extracted in dimethyl sulfoxide (DMSO), supplemented with deuteroporphyrin as an internal standard, and analyzed by LC-MS/MS. Integrated analyte peak areas were normalized to PPIX in each sample. RBC: uninfected red blood cells, WT: parental clone 3D7, IPP/dox: isopentenyl pyrophosphate/doxycycline-treated 3D7 parasites. (**B**, **C**) Growth of asynchronous ΔPBGD (**B**) and ΔCPO (**C**) 3D7 parasites in the presence or absence of 200 μM ALA, with 2-min light exposures on an overhead projector on days 0–2. WT growth was fit to an exponential equation. (**D**) Bright field and fluorescence images of live 3D7 parasites expressing ALAD-GFP from a plasmid before or after 2-week treatment with IPP and doxycycline. (**E**) Growth of asynchronous IPP/doxycycline-treated parasites in the presence or absence of 200 μM ALA and 50 μM SA, with 2-min light exposures on an overhead projector on days 0–2.

The following figure supplements are available for figure 2:

**Figure supplement 1**. Immunofluorescence microscopy (IFM) images of fixed 3D7 parasites expressing full-length PBGD tagged at its endogenous locus with C-terminal GFP confirm targeting of the native protein to the parasite apicoplast.

**Figure supplement 2**. Fluorescence microscopy images of live 3D7 parasites episomally expressing full-length CPO with a C-terminal GFP tag confirm protein localization to the parasite cytoplasm.

**Figure supplement 3**. Disruption of the *P. falciparum* PBGD gene (PF3D7_1209600) by single-crossover homologous recombination.

**Figure supplement 4**. Disruption of the *P. falciparum* CPO gene (PF3D7_1142400) by single-crossover homologous recombination.

**Figure supplement 5**. IFM images of fixed 3D7 parasites expressing full-length ALAD tagged at its endogenous locus with C-terminal GFP confirm targeting to the parasite apicoplast.

**Figure supplement 6**. PCR analysis of genomic DNA from untreated WT parasites or parasites cultures ≥7 days in 1 μM doxycycline and 200 μM IPP.

**Figure supplement 7**. Western blot analysis of parasites episomally expressing ALAD-GFP and cultured 7 days in IPP and doxycycline.

thought to be predominantly targeted to the mitochondrial intermembrane space (IMS) (*Ponka, 1997*; *van Dooren et al., 2012*), persists in the erythrocyte cytoplasm after maturation of precursor reticulocytes and mitochondrial loss. Indeed, other mitochondrial IMS proteins, such as cytochrome c, are known to partition into the cytoplasm under certain conditions (*Liu et al., 1996*; *Soltys and Gupta, 2000*). CPO catalyzes production of protoporphyrinogen IX, which in the oxygen-rich environment of erythrocytes can spontaneously oxidize to form PPIX (*Wang et al., 2008*).

In support of this model, we noted that red porphyrin fluorescence in ALA-treated infected erythrocytes was not limited to the parasite but was detectable throughout the erythrocyte cytoplasm (*Figure 1B*), as expected for host enzyme activity and production of PPIX in this compartment. To directly test the model that enzymes remaining in the erythrocyte cytoplasm could catalyze PPIX biosynthesis from ALA, we permeabilized uninfected erythrocytes using the detergent saponin, clarified lysates by centrifugation followed by sterile filtration, and then used LC-MS/MS to monitor heme and porphyrin biosynthesis from 5-[$^{13}C_4$]-ALA added to the filtered lysate supernatant. We detected formation of $^{13}$C-labelled PPIX and CPP but not heme, and this biosynthetic activity was fully blocked by SA (*Figure 3A*). These observations provide direct support for our model that erythrocytes retain a vestigial and partial biosynthesis pathway capable of converting exogenous ALA into PPIX but are unable to convert PPIX to heme due to lack of mitochondria and FC.

## ALA uptake by erythrocytes requires new permeability pathways established by *Plasmodium* infection

In contrast to parasite-infected erythrocytes, uninfected erythrocytes showed no detectable porphyrin fluorescence in the presence of ALA (*Figure 3B*), consistent with reports that the erythrocyte membrane has a low permeability to amino acids (*Ginsburg et al., 1985*; *Kirk et al., 1994*; *Desai et al., 2000*). Electroporation of uninfected erythrocytes in the presence of ALA, however, resulted in robust intracellular porphyrin fluorescence (*Figure 3B*), supporting our model that host enzymes have

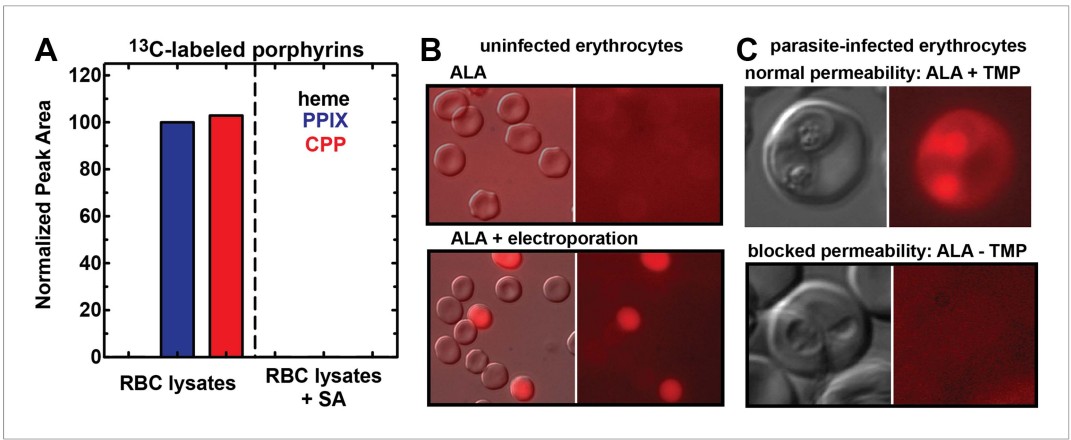

**Figure 3**. Erythrocytes have latent porphyrin biosynthesis activity that requires exogenous ALA and parasite permeability mechanisms to enable ALA uptake. (**A**) LC-MS/MS detection of $^{13}$C-labelled PPIX and CPP in erythrocyte lysate supernatants incubated with 200 µM 5-[$^{13}C_4$]-ALA without or with 50 µM SA. Erythrocytes were lysed in 0.04% saponin, centrifuged at 25,000×g for 60 min, and 0.2 µM syringe filtered prior to ALA addition. (**B**) Bright field and fluorescence (Zeiss filter set 43 HE) images of uninfected erythrocytes incubated in 500 µM ALA before or after electroporation. (**C**) Bright-field and fluorescence images of parasites cultured in 500 µM ALA with normal (+TMP) or blocked (−TMP) establishment of parasite permeability pathways in the erythrocyte membrane. Infected erythrocyte permeability was modulated using a 3D7 parasite line expressing HSP101 tagged at its endogenous locus with a TMP-dependent destabilization domain (*Beck et al., 2014*). TMP was maintained or washed out from synchronous schizont-stage parasites, which were allowed to rupture and invade new erythrocytes. 500 µM ALA was added to both cultures after invasion, and parasites were imaged 8 hr later.

The following figure supplement is available for figure 3:

**Figure supplement 1**. Furosemide blocks ALA uptake and PPIX biosynthesis in parasite-infected erythrocytes.

latent biosynthetic activity that requires a mechanism for ALA uptake across the normally impermeable erythrocyte membrane.

Upon invasion, *Plasmodium* parasites export hundreds of effector proteins into host erythrocytes. These proteins dramatically alter the architecture of the infected erythrocyte and establish new permeability pathways (NPPs) that enhance host cell uptake of amino acids and other nutrients from host serum (*Spillman et al., 2015*). We hypothesized that selective uptake of ALA by parasite-infected erythrocytes was mediated by NPP mechanisms.

To test this model, we utilized a recently published parasite line in which the export of parasite proteins into host erythrocytes, including establishment of nutrient permeability pathways, can be conditionally regulated with the synthetic small-molecule ligand trimethoprim (TMP) (*Beck et al., 2014*). In these parasites, protein export and NPP mechanisms are functional in the presence of TMP but are blocked in its absence. We maintained or washed out TMP from a synchronized culture of late schizonts, allowed parasites to rupture and reinvade new erythrocytes, and then incubated both sets of parasites in ALA for 8 hr. Whereas TMP-treated parasites with normal protein export and permeability displayed robust PPIX fluorescence indistinguishable from that of wild-type parasites, parasites incubated without TMP showed no detectable PPIX fluorescence (*Figure 3C*). We observed similar results in wild-type parasites using the small-molecule drug furosemide, which blocks parasite-derived NPP mechanisms directly (*Figure 3—figure supplement 1*) (*Staines et al., 2004*). We conclude that ALA is selectively taken up by infected erythrocytes via parasite-dependent nutrient-acquisition pathways and metabolized to PPIX by vestigial host enzymes within the erythrocyte cytoplasm (*Figure 4*).

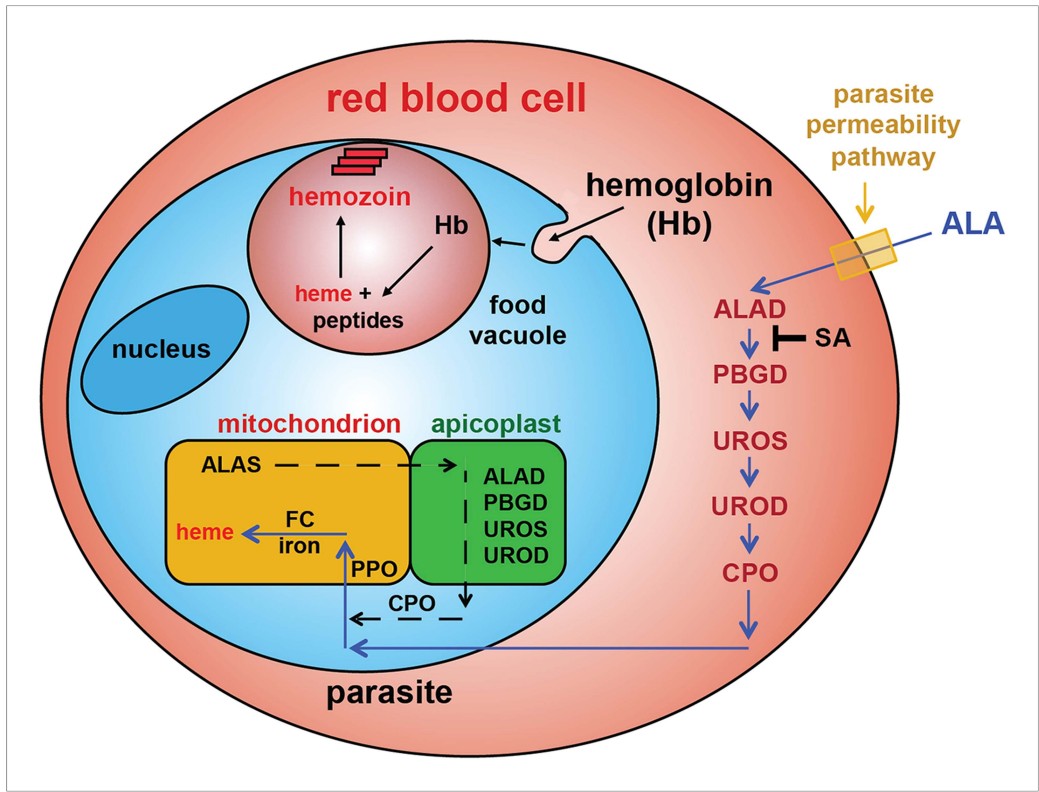

**Figure 4**. Schematic depiction of ALA-uptake and porphyrin biosynthesis pathways in Plasmodium-infected erythrocytes. For simplicity, all membranes are depicted as single. Porphyrins synthesized in the infected erythrocyte cytoplasm from exogenous ALA may be transported across the parasite membrane via unspecified mechanisms or may be taken up via hemoglobin import mechanisms. SA inhibits ALAD and blocks porphyrin synthesis from ALA.

The following figure supplement is available for figure 4:

**Figure supplement 1**. Growth of parasites in 1 μM 5-[$^{13}C_4$]-ALA results in detectable biosynthesis of $^{13}$C-labeled PPIX.

## The *Plasmodium*-encoded heme biosynthesis pathway has no detectable activity during blood-stage infection

Our observation that disruption of the parasite-encoded PBGD and CPO does not affect biosynthetic production of heme and PPIX from ALA suggests that host enzyme activity in the erythrocyte cytoplasm provides the dominant contribution to PPIX biosynthesis in ALA-treated intraerythrocytic parasites. To dissect these parallel pathways and test whether the parasite pathway alone, in the absence of host enzymes, can support heme biosynthesis from ALA, we fractionated parasite-infected erythrocytes using saponin to selectively permeabilize host cell membranes while leaving parasite membranes intact (*Figure 5A*). Under these conditions, soluble erythrocyte proteins can be washed away to leave the intact live parasite natively embedded within the resulting erythrocyte ghost (*Figure 5B,C*). Parasites treated in this fashion remain metabolically active for 5–6 hr or longer, retain a membrane potential, accumulate fluorescent dyes such as MitoTracker Red (*Figure 5B,C*), and carry out DNA synthesis (*Izumo et al., 1987*; *Cobbold et al., 2011*). After saponin treatment and washout, we placed parasites back into culture medium containing $^{13}$C-ALA and incubated them overnight before extracting them for analysis by LC-MS/MS. We failed to detect biosynthesis of heme, PPIX, or CPP in fractionated asexual and sexual blood-stage parasites (*Figure 5D*), suggesting that the apicoplast-localized portion of the parasite heme biosynthesis pathway is largely or completely inactive during blood-stage infection.

To test this conclusion within undisrupted parasites, we created a transgenic parasite line expressing an apicoplast-targeted version of the uroporphyrinogen III methyltransferase (cobA) protein from *Propionibacterium freudenreichii* to serve as a biosensor of heme biosynthesis pathway activity within the parasite apicoplast. The cobA protein catalyzes conversion of the heme biosynthesis intermediate uroporphyrinogen III into the red fluorescent products sirohydrochlorin and trimethylpyrrocorphin (*Figure 6A*) and has been shown to function when heterologously expressed in bacteria (*Figure 6B*), yeast, and mammalian cells (*Sattler et al., 1995*; *Wildt and Deuschle, 1999*). We targeted a cobA-green fluorescent protein (GFP) fusion protein to the apicoplast in ALA-treated parasites (*Figure 6C*) but were unable to detect any red fluorescence in this organelle indicative of cobA-mediated conversion of uroporphyrinogen III to the fluorescent products. This result suggests that parasite enzymes targeted to

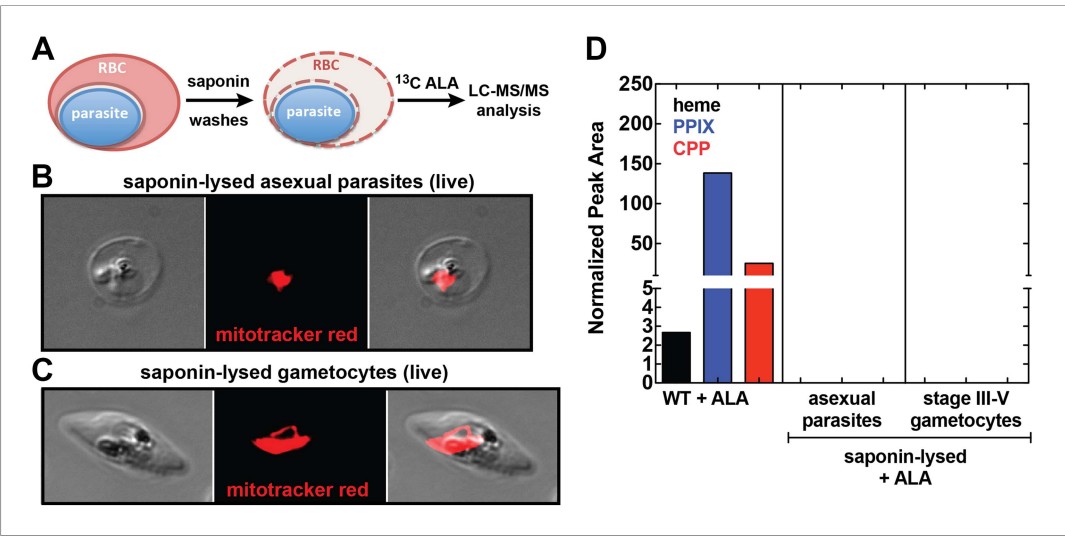

**Figure 5**. Analysis of heme biosynthesis activity in parasite-infected erythrocytes after saponin permeabilization and culture in $^{13}$C-labelled ALA. (**A**) Parasite-infected erythrocytes were permeabilized in 0.02% saponin, washed to remove the erythrocyte cytoplasm, and placed back into culture medium containing 200 µM 5-[$^{13}$C$_4$]-ALA for 12 hr prior to DMSO extraction and analysis by LC-MS/MS. Bright field and fluorescence image of live (**B**) asexual trophozoite and (**C**) stage IV sexual gametocyte treated with 0.02% saponin and stained with 20 nM MitoTracker Red. (**D**) LC-MS/MS quantification of $^{13}$C-labelled heme, PPIX, and CPP in DMSO extracts of intact WT 3D7 asexual parasites, saponin-released asexual parasites, and saponin-released gametocytes cultured overnight in 200 µM 5-[$^{13}$C$_4$]-ALA.

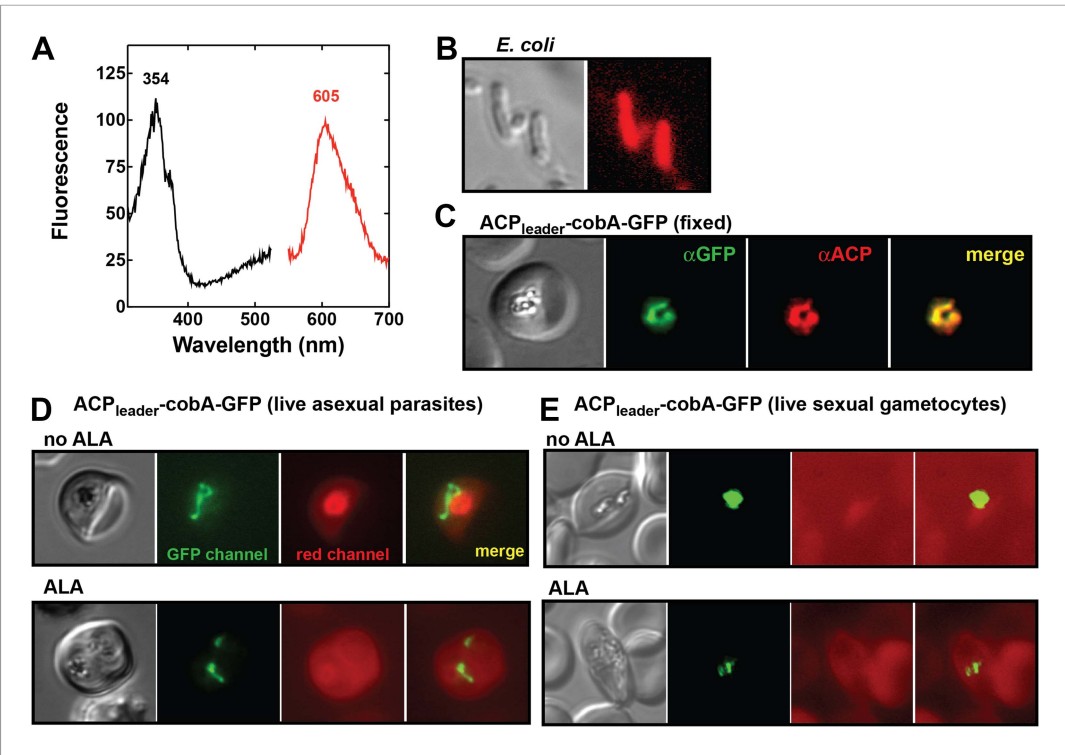

**Figure 6**. Analysis of heme biosynthetic flux within the apicoplast of live parasites using the cobA biosensor. (**A**) Fluorescence excitation (black) and emission (red) spectra of clarified lysates from *Escherichia coli* bacteria expressing the cobA gene from *Propionibacterium freudenreichii* (*shermanii*), showing the expected peaks for conversion of uroporphyrinogen III to sirohydrochlorin and trimethylpyrrocorphin. (**B**) Fluorescence microscopy images of live bacteria expressing the cobA gene, acquired on the bright field and red (Zeiss filter set 43 HE) channels. (**C**) Immunofluorescence (IFM) images of fixed 3D7 parasites episomally expressing an ACP$_{leader}$-cobA-GFP fusion confirm targeting to the parasite apicoplast. Parasites were stained with αGFP and αACP (acyl carrier protein [ACP], apicoplast marker). The αACP antibody recognizes an epitope that is different from the ACP leader sequence. (**D**) Fluorescence microscopy images of live asexual parasites episomally expressing ACP$_{leader}$-cobA-GFP without or with 500 µM exogenous ALA. (**E**) Fluorescence microscopy images of live stage III–IV sexual gametocytes episomally expressing ACP$_{leader}$-cobA-GFP without or with 500 µM exogenous ALA. Fluorescence images in (**D**) and (**E**) were acquired on the GFP (Zeiss filter set 38) and red (Zeiss filter set 43 HE) channels.

The following figure supplement is available for figure 6:

**Figure supplement 1**. Disruption of the *Plasmodium* FC gene in ΔFC parasites does not photosensitize parasites.

this organelle are inactive in both asexual and sexual blood stages (*Figure 6D,E*). We also noted that growth of ΔFC parasites (*Ke et al., 2014*), which would be expected to accumulate PPIX during native pathway activity (*Figure 1A*), was insensitive to light in the absence of ALA (*Figure 6—figure supplement 1*). These observations suggest that heme biosynthesis in blood-stage *P. falciparum* parasites is only operative when exogenous ALA is present to stimulate PPIX production by remnant host enzymes in the erythrocyte cytoplasm, with subsequent PPIX uptake and conversion to heme by FC in the parasite mitochondrion (*Figure 4*).

## Development of chemiluminescence-based PDT for treatment of blood-stage malaria

Our results and those of prior studies suggested that stimulation of heme biosynthesis might be exploited for antimalarial PDT. Conventional PDT requires an external light source to photoactivate and kill ALA-stimulated cells (*Juzeniene et al., 2007*; *Celli et al., 2010*; *Wachowska et al., 2011*). While such approaches can successfully target localized shallow-tissue tumors (*Celli et al., 2010*;

*Wachowska et al., 2011*), they are impractical for treating malaria due to the dispersed nature of blood-stage infection, the sequestration of mature *P. falciparum* parasites along the vascular endothelium, and the requirement to illuminate every infected erythrocyte.

To bypass the need for external light, chemiluminescence has been proposed as an alternative means to photoactivate the cytotoxicity of PPIX in ALA-stimulated cells. Trial studies in cancer cell lines have shown modest success using the small molecule luminol (*Laptev et al., 2006*; *Chen et al., 2012*), whose iron-activated chemiluminescence (*Rose and Waite, 2001*) overlaps the absorbance spectrum of PPIX (*Figure 7—figure supplement 1*). Cancer cells tightly sequester iron, which may limit luminol activation in these environments. *Plasmodium* parasites, however, expose abundant iron during large-scale digestion of erythrocyte hemoglobin and liberation of heme. We therefore hypothesized that intraerythrocytic parasites might show heightened susceptibility to chemiluminescence-based PDT (CL-PDT) (*Figure 7A*).

To test the efficacy of a CL-PDT strategy for targeting blood-stage *P. falciparum*, we incubated intraerythrocytic parasites with twice-daily media changes in combinatorial treatments of ALA, luminol, and 4-iodophenol, a small molecule that has been shown to enhance the intensity and duration of luminol chemiluminescence (*Thorpe et al., 1985*). Parasites treated with optimized concentrations of each compound in single or double combination showed little effect on parasite growth (*Figure 7—figure supplement 2*). The combination of all three compounds, however,

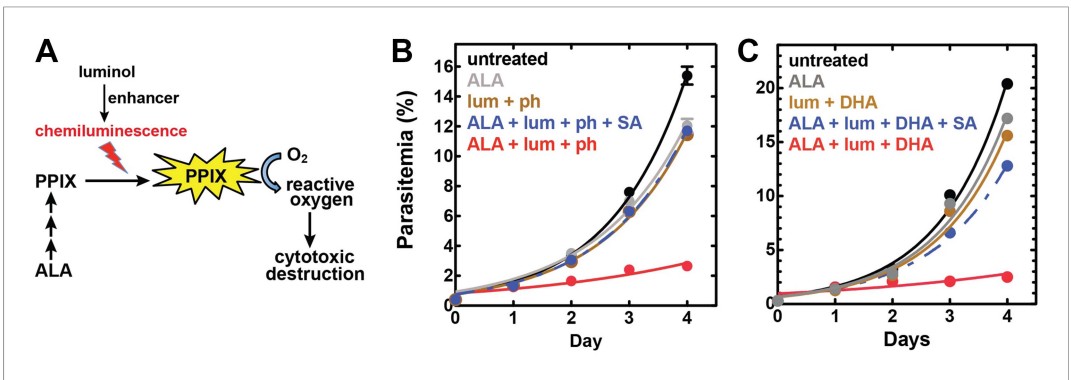

**Figure 7**. Targeting blood-stage Plasmodium parasites by chemiluminescence-based photodynamic therapy (CL-PDT). (**A**) Schematic depiction of a CL-PDT mechanism for targeting blood-stage malaria. (**B**) Effect of 100 μM ALA, 750 μM luminol (lum), 50 μM 4-iodophenol (ph), 50 μM SA and their combination (all 0.25% DMSO) on the growth of asynchronous 3D7 parasites. (**C**) Effect of 100 μM ALA, 750 μM luminol, 0.5 nM dihydroartemisinin (DHA), 50 μM SA and their combination (all 0.25% DMSO) on the growth of asynchronous 3D7 parasites. Parasite media was changed twice daily, and parasitemia increases were fit to an exponential growth equation.

The following figure supplements are available for figure 7:

**Figure supplement 1**. Spectral compatibility of luminol and PPIX.

**Figure supplement 2**. Effect of combinatorial ALA, luminol, and 4-iodophenol treatment on parasite growth.

**Figure supplement 3**. Giemsa-stained blood smear of 3D7 parasite culture after 3 days of treatment in 100 μM ALA, 750 μM luminol, and 50 μM 4-iodophenol.

**Figure supplement 4**. Efficacy of CL-PDT with drug-resistant parasites.

**Figure supplement 5**. Concentration dependence of growth inhibition of 3D7 parasites by DHA over 48 hr (in 0.2% DMSO).

**Figure supplement 6**. Effect of 500 pM DHA, 100 μM ALA, and their combination (all in 0.2% DMSO) on the growth of asynchronous 3D7 parasites.

potently inhibited parasite growth (*Figure 7B*), and microscopic examination by blood smear revealed widespread parasite death (*Figure 7—figure supplement 3*). The growth effects of this combination could be fully rescued by SA (*Figure 7B*), supporting a photodynamic mechanism requiring PPIX biosynthesis from ALA and enhanced chemiluminescence by luminol.

The development of parasite resistance to frontline antimalarial drugs continues to hamper malaria treatment and eradication efforts worldwide. To test whether a CL-PDT mechanism remains effective against parasites with diverse resistance to distinct drugs, we turned to studies with Dd2 parasites, which have multidrug resistance to antifolate and quinolone antibiotics (*Sidhu et al., 2002*), and a clinical isolate containing a kelch-13 protein mutation that confers artemisinin tolerance (*Ariey et al., 2014*). In both parasite lines, combination treatment with ALA, luminol, and 4-iodophenol potently ablated parasite growth (*Figure 7—figure supplement 4*).

Artemisinin and its derivatives are current frontline drugs used for treatment of malaria, usually in combination with a partner drug. Artemisinin, whose endoperoxide moiety is reductively cleaved by intracellular iron to generate reactive oxygen radicals (*Meshnick, 2002*), has been shown to potently stimulate luminol chemiluminescence in vitro (*Green et al., 1995*), suggesting the possibility of combining CL-PDT with artemisinin for antimalarial treatment. To test the efficacy of an artemisinin-enhanced CL-PDT strategy, we incubated drug-sensitive 3D7 parasites with twice-daily media changes in ALA, luminol, and sub-therapeutic doses (500 pM, 10% of $EC_{50}$, *Figure 7—figure supplement 5*) of dihydroartemisinin (DHA). Single and double combinations had little effect (*Figure 7—figure supplement 6*), but all three compounds together potently ablated parasite growth, and growth inhibition could be fully rescued with SA (*Figure 7C*).

## Discussion

Malaria parasites, like nearly all other organisms, have a metabolic requirement for heme (*Painter et al., 2007*; *van Dooren et al., 2012*; *Sigala and Goldberg, 2014*). Despite access to abundant host heme during intraerythrocytic infection, parasites retain a complete heme biosynthesis pathway that was regarded for two decades as essential and a potential drug target (*Surolia and Padmanaban, 1992*). Our work and recent studies have now successfully disrupted four of the eight pathway enzymes, providing firm evidence that de novo heme synthesis is dispensable during blood-stage infection. These data strongly suggest that parasites have mechanisms to scavenge host heme to meet metabolic needs and clarify that heme biosynthesis is not a viable target for classical drug inhibition (*Nagaraj et al., 2013*; *Ke et al., 2014*).

Our results further suggest that the parasite pathway is not active during intraerythrocytic infection. This inactivity may reflect the low availability of the succinyl-CoA precursor due to limited tricarboxylic acid cycle flux (*MacRae et al., 2013*) and additional regulatory mechanisms that suppress the activity of apicoplast-targeted enzymes during blood-stage parasite development, such as feedback inhibition by host heme or active-site inhibition by endogenous metabolites (*Park et al., 2015*),. The functional quiescence of this specific biosynthetic pathway, despite expression of the constituent enzymes, may reflect a general survival strategy and adaptation by parasites to closely match their metabolic requirements with the nutritional availability of specific host environments, such as those previously described for parasite acquisition of fatty acids (*Yu et al., 2008*). Indeed, prior studies suggest that the parasite heme synthesis pathway is required for development within the mosquito and human liver (*Nagaraj et al., 2013*; *Ke et al., 2014*). These stages involve host environments with lower heme availability compared with erythrocytes and in which parasite heme requirements appear to be elevated due to enhanced reliance on mitochondrial electron transport for ATP synthesis (*van Dooren et al., 2012*; *Sigala and Goldberg, 2014*; *Sturm et al., 2015*). A prior study reported that blood-stage *P. falciparum* parasites adopt distinct physiological states in vivo, including a state with heightened oxidative metabolism and mitochondrial activity that may arise during host starvation (*Daily et al., 2007*). It remains a future challenge to test whether heme biosynthesis by the parasite-encoded pathway can be stimulated by host nutritional status during intraerythrocytic infection.

Despite the dispensable nature and apparent inactivity of the parasite-encoded heme biosynthesis pathway, infected erythrocytes retain a paradoxical ability to synthesize heme from exogenous ALA. This biosynthetic activity requires the *Plasmodium* FC but not the upstream enzymes in the parasite pathway. Indeed, knock-out of the *Plasmodium* FC prevented conversion of PPIX to heme (*Ke et al., 2014*), but our disruption of the parasite PBGD and CPO genes, as well as the entire apicoplast

organelle, had no effect on ALA-stimulated heme synthesis (*Figure 2*). Our work resolves this paradox by identifying a latent contribution to heme biosynthesis in parasite-infected erythrocytes from vestigial host enzymes remaining in the erythrocyte cytoplasm. Since erythrocytes lack mitochondria, they are missing the initial and terminal pathway enzymes, are unable to synthesize ALA, and thus retain only a truncated and normally inactive heme synthesis pathway. Analytical studies have reported that human serum contains 0.1–0.2 µM ALA (*Zhang et al., 2011*), but the low permeability of the erythrocyte membrane to ALA means that extracellular ALA is largely inaccessible to vestigial host enzymes within uninfected erythrocytes.

Our study clarifies that NPP mechanisms of parasite-infected red blood cells enable efficient uptake of exogenous ALA. After uptake, ALA can be metabolized by vestigial human enzymes within the oxygen-rich erythrocyte cytoplasm to produce PPIX that can be converted to heme by FC within the parasite mitochondrion (*Figure 4*). Thus, the 0.2 µM ALA natively present in human serum may be sufficient to stimulate low-level heme biosynthesis within parasites in vivo. Indeed, we detect $^{13}$C-labeled PPIX in parasites grown in 1 µM 5-[$^{13}$C$_4$]-ALA in vitro (*Figure 4—figure supplementary 1*), supporting a model that serum ALA stimulates heme biosynthesis during malaria infection in vivo. Such activity, however, is not required to support blood-stage parasite growth. We note that prior work suggested a role for remnant host enzymes in heme biosynthesis by malaria parasites, but this prior proposal differed by positing that human enzymes such as ALAD were somehow imported and active within the parasite cytoplasm (*Padmanaban et al., 2007*).

The ability to photosensitize parasites with exogenous ALA and then kill them with light introduces exciting possibilities for developing new photodynamic treatment strategies, exemplifies how a deep understanding of fundamental parasite metabolism can be leveraged for designing novel therapies, and highlights that non-essential pathways can still serve as therapeutic targets. We have shown that luminol-based chemiluminescence, when stimulated by combinatorial delivery of low-dose 4-iodophenol or artemisinin, can circumvent the conventional PDT requirement for external light and potently ablate parasite growth (*Figure 7*). We note that ALA, luminol, and DHA have excellent toxicity profiles, favorable pharmacokinetic properties, and have each been used clinically (*Bissonnette et al., 2001*; *Dalton et al., 2002*; *Gordi and Lepist, 2004*; *Larkin and Gannicliffe, 2008*; *Wachowska et al., 2011*). These results suggest the possibility of including ALA and luminol with therapeutic doses of artemisinin (or its clinical derivatives) as a novel form of artemisinin combination therapy for treating malaria. Multidrug-resistant parasites remain susceptible to this photodynamic strategy, which relies on host enzyme activity outside the genetic control of parasites and thus is refractory to the development of resistance-conferring mutations. This CL-PDT strategy may also be effective against other intraerythrocytic parasites, such as *Babesia*. As for any new therapy suggested by in vitro studies, additional in vivo experiments will be required to optimize treatment regimens for our proposed therapy and to confirm efficacy and safety.

Finally, we note that this strategy of using artemisinin to stimulate intracellular light emission by luminol for ALA-based PDT may be applicable to treating deep-tissue cancers, for which poor accessibility to external light also limits current PDT approaches in development for cancer therapy (*Celli et al., 2010*). Artemisinin has shown promising anti-cancer properties on its own (*Nakase et al., 2008*), and thus its combination with ALA and luminol may provide potent synergy for cancer chemotherapy.

## Materials and methods

### Materials

All reagents were of the highest purity commercially available. SA, 5-ALA, 4-iodophenol, TMP, furosemide, saponin, IPP, doxycycline, luminol, and DHA were purchased from Sigma (St. Louis, MO, USA). 5-[$^{13}$C$_4$]-ALA was purchased from Cambridge Isotope Laboratories, Inc (Tewksbury, MA, USA).

### Microscopy

Images of live or fixed parasites were acquired on an Axio Imager M1 epifluorescence microscope (Carl Zeiss Microimaging, Inc.) equipped with a Hamamatsu ORCA-ER digital CCD camera and running Axiovision 4.8 software, as described previously (*Sigala et al., 2012*). Live parasite nuclei were stained with 5 µM Hoechst 3342 added immediately prior to image acquisition. For photodynamic imaging studies, parasites were cultured in 200–500 µM ALA in the absence or presence of 50 µM SA for 6–12 hr prior to visualization. Hemozoin movement in parasite digestive vacuoles was imaged by acquiring 10–20

sequential frames at 1 s intervals on the bright-field channel. Images were cropped and superimposed in Adobe Photoshop, exported as video files with a 0.1 s frame delay, and looped for 15 seconds of playback. Immunofluorescence (IFM) images were acquired by fixing and staining parasites as previously described (Tonkin et al., 2004; Ponpuak et al., 2007). Electron microscopy images of parasites subjected to light or 500 μM ALA + light were obtained as previously described (Beck et al., 2014). Uninfected erythrocytes were washed and resuspended in 1× cytomix containing 500 μM ALA, electroporated in a manner identical to parasite transfections (see below), washed in phosphate-buffered saline (PBS), incubated overnight at 37°C, and imaged as described above for live parasites. Images acquired on different channels for common samples were processed with identical brightness and contrast settings.

## Parasite growth analysis

Parasite growth was monitored by diluting asynchronous parasites to 0.5% parasitemia and allowing culture expansion with daily or twice-daily media changes. Parasitemia was measured daily by diluting 10 μl of each resuspended culture in 200 μl acridine orange (1.5 μg/ml) and analyzing by flow cytometry, as previously described (Muralidharan et al., 2012). To assess the light sensitivity of ALA-treated parasites, asynchronous parasites were cultured in 200 μM ALA in the absence or presence of 50 μM SA and subjected to 2 min daily exposures to broad-wavelength white light on an overhead projector. Daily parasitemia measurements were plotted as a function of time and fit to an exponential growth equation using GraphPad Prism 5.0.

For chemiluminescence experiments, asynchronous parasites were diluted to 0.5% parasitemia and incubated in $\pm$100 μM ALA and $\pm$50 μM SA for 8 hr. After 8 hr, parasite media was changed to also include 750 μM luminol, 50 μM 4-iodophenol, and 0.5 nM DHA in the indicated combinations. Parasite cultures were allowed to expand over 5 days, with twice-daily (7 am and 4 pm) media changes in the indicated combinations. Parasitemia was measured daily on replicate samples, as indicated above. Experiments were performed using 3D7 (drug sensitive) parasites, Dd2 (multidrug resistant) parasites (Sidhu et al., 2002), and a clinical isolate (Ariey et al., 2014) (MRA-1241) bearing the I543T mutation in the Kelch-13 gene locus responsible for artemisinin tolerance. Exposure of parasite cultures to ambient light was minimized by changing media within darkened TC hoods and covering parasite culture dishes during brief (5–10 s) transits to and from incubators. The effect of 100 μM ALA on parasite growth varied slightly between experiments, possibly due to differences between distinct batches of donated erythrocytes.

For $IC_{50}$ determinations, asynchronous parasites were diluted to 1% parasitemia and incubated with variable drug concentrations for 48 hr without media change. After 48 hr, parasitemia was determined in duplicate samples for each drug concentration, normalized to the parasitemia in the absence of drug, plotted as a function of the log of the drug concentration, and fit to a sigmoidal growth inhibition curve using GraphPad Prism 5.0.

## Parasite strains, culture, genetic modification, and transgene expression

Parasite culture and transfection were performed in Roswell Park Memorial Institute medium (RPMI) supplemented with Albumax, as previously described (Sigala et al., 2012). Cloning was performed using either restriction endonuclease digestion and ligation or the In-Fusion system (Clontech Mountain View, CA, USA).

For episomal expression of *P. falciparum* ALA dehydratase (PF3D7_1440300) and CPO (PF3D7_1142400) fused to a C-terminal GFP tag (ALAD-GFP and CPO-GFP), cDNA inserts encoding the complete ALAD and CPO genes (exons only) were reverse transcription (RT)-PCR amplified from total parasite RNA using the Superscript III system (Life Technologies) and primers CACTATA GAACTCGAGATGTTAAAATCAGATGTA GTGCTTTTATTGTATATAC and CTGCACCTGGCCTAGG TAGAGTTAATTCTATATT AAAATTATTATTTGAATTATCATC (ALAD) or ACGATTTTTTCTCGAGAT GAAA GATGAGATAGCTCCTAATGAATATTTTAGAAATTTATG and CTGCACCTGGCCT AGGG TAGTCCACCCACTTTTTGGGATAC (CPO). These were digested with XhoI/AvrII and ligated into the XhoI/AvrII sites of a digested pTEOE vector that was identical to a previously described pTyEOE vector (Beck et al., 2014) except that the pTEOE plasmid contained human dihydrofolate reductase (DHFR) in place of yeast dihydroorotate dehydrogenase (yDHOD) as the positive selection marker. Plasmid-based expression of the ALAD–GFP fusion was driven by the HSP86 promoter. This plasmid

(50 µg) was co-transfected into 3D7 parasites along with plasmid pHTH (10 µg) for transient expression of the *piggyback* transposase that mediates integration of the pTEOE plasmid into the parasite genome (*Balu et al., 2005*). Parasites were selected with 5 nM WR99210.

For episomal expression of *P. freudenreichii* (*shermanii*) uroporphyrinogen III methyltransferase (cobA) (Genbank: CBL55989.1) targeted to the parasite apicoplast, the cobA gene was PCR amplified from plasmid pISA417 (*Sattler et al., 1995*) using the primers ACGATTTTTTCTCGAGATGACCAC CACACTGTTGCCCGGCACTGTC and CTGCACCT GGCCTAGGGTGGTCGCTGGGCGCGCGATGG, digested with XhoI/AvrII and ligated into the XhoI/AvrII-cut pTEOE vector described above. An insert encoding the *P. falciparum* acyl carrier protein (ACP) leader sequence (residues 1–60) with 5′- and 3′-XhoI sites, previously PCR amplified from parasite cDNA (*Sigala et al., 2012*), was digested with XhoI and ligated into the XhoI-cut cobA/pTEOE vector to generate an in-frame ACP$_L$-cobA-GFP fusion gene. This plasmid was co-transfected with pHTH into 3D7 parasites as described above.

For disruption of the *P. falciparum* genes encoding PBGD (PF3D7_1209600) and CPO (PF3D7_1142400), primer pairs (PBGD: CACTATAGAACTCGAGGATCATAATAA TGATACATTATG TACTATTGGGACATCGTCC and CTGCACCTGGCCTAGGAA CTGCTATAATGCCTTGACCTAAGGC AGGATAAATCAGG; CPO: CACTATAG AACTCGAGTTTTTTCAAATATTTATAAAAACAGGAAAAA AGAAGAAAAAATA and CTGCACCTGGCCTAGGATAACATTTACAATCCTTATTATTATTATTATTAT TGTTG ATGG) were used to PCR amplify 360 bp and 471 bp sequences from the middle of the 1.3 kb PBGD and 1.6 kb CPO genes, respectively. These inserts were cloned by In-Fusion into the XhoI/AvrII sites of the pPM2GT vector (*Klemba et al., 2004*), which encodes a C-terminal GFP tag after the AvrII site and also contains a human DHFR marker for positive selection with 5 nM WR99210. This vector was further modified to introduce a 2A peptide sequence (*Straimer et al., 2012*) followed by the yeast DHOD sequence (*Ganesan et al., 2011*), after the 3′ end of the GFP cassette, to enable positive selection for integration with the parasite DHOD inhibitor DSM1. The yDHOD marker, however, was not used for selection in this study, and use of the GFP-2A-yDHOD/PM2GT vector for positive selection of plasmid integration into the genome will be described elsewhere. Plasmids (50 µg) were transfected into 3D7 parasites by electroporation. Parasites were subjected to three rounds of positive selection with 5 nM WR99210. After the first and second selections, cultures were maintained for 3 weeks in the absence of drug pressure prior to the subsequent round of positive selection. After the third round of positive selection, parasites were cloned by limiting dilution. Clonal parasites that had integrated the plasmid at the desired locus to disrupt the target genes by single cross-over homologous recombination were verified by PCR and Southern blot, as previously described (*Klemba et al., 2004*), and retained for further analysis.

To introduce a C-terminal GFP tag into the endogenous locus of the *P. falciparum* PBGD gene (PF3D7_1209600), primer pairs CACTATAGAACTCGAGGATCATAATAA TGATACATTATGTACTAT TGGGACATCGTCC and CTGCACCTGGCCTAGGTTTATTA TTTAAAAGGTGCAATTCAGCCTCCGC TTTTATTTTG were used to PCR amplify the complete PBGD coding sequence. This insert was cloned by In-Fusion into the 2A-yDHOD/PM2GT vector described above and transfected into 3D7 parasites by electroporation as above. Stable integration was achieved after three rounds of positive selection in WR99210, and clones were isolated by limiting dilution and verified by PCR and Southern blot.

ΔFC D10 parasites, described in a prior study (*Ke et al., 2014*), were obtained from Akhil Vaidya (Drexel University) and were cultured as described above, including 2-min daily exposures to broad-wavelength white light using an overhead projector light box.

For apicoplast disruption experiments, parasites were cultured in 1 µM doxycycline and 200 µM IPP for 7–21 days (*Yeh and DeRisi, 2011*). After 7 days, genomic DNA was harvested and analyzed by PCR for the nuclear-encoded ACP gene (primers ATGAAGATCTTATTACTTTGTATAATTTTTC and TTTTAAAGAGCTAGATGGGTTTTT ATTTTTTATC) and apicoplast-encoded rps8 (primers ATGATTAT TAAATTTTTAAATAATG and TTACAAAATATAAAATAATAAAATACC) and ORF91 (primers ATG ACTTT ATATTTAAATAAAAATTT and TTACATATTTTTTTTTATTGAAGAACG) genes to confirm selective loss of the apicoplast genome.

## Gametocyte induction and culturing

3D7 ΔPM1 (plasmepsin 1) parasites (*Liu et al., 2005*) and 3D7 parasites expressing ACP$_{leader}$-cobA-GFP from a pTEOE plasmid (described above) were synchronized using 5% sorbitol and cultured using gentamycin-free media. Gametocytogenesis was stress-induced in mid-trophozoite parasites (5–7%

parasitemia) by increasing hematocrit to 4% (by removing half of the culture medium volume) for 12 hr. Parasites were maintained in culture for 4–6 days, at which point mostly stage III–IV gametocytes were visible. ACP$_{leader}$-cobA-GFP gametocytes were incubated overnight in the presence or absence of 500 µM ALA and imaged by live parasite microscopy as described above. For $^{13}$C-ALA labeling experiments, gametocytes were passaged over a magnetic column to remove uninfected erythrocytes and concentrate the gametocyte-infected cells, lysed in 0.02% saponin (0.2-µM filtered), placed back into culture medium containing 200 µM 5-[$^{13}$C$_4$]-ALA, and incubated overnight at 37°C. Parasites were then isolated by centrifugation, extracted, and analyzed by tandem mass spectroscopy as described below.

## Analysis of heme biosynthesis by $^{13}$C-labeling and LC-MS/MS

Parasites were cultured in 200 µM 5-[$^{13}$C$_4$]-ALA for 12–24 hr, harvested by centrifugation, lysed in 0.05% cold saponin, washed in PBS, and extracted with dimethyl sulfoxide (DMSO). Deuteroporphyrin was added as an internal standard and extracts were analyzed for $^{13}$C-labeled heme, PPIX, and coproporphyrin III (CPPIII) using a previously published LC-MS/MS assay (*Ke et al., 2014*). Detection of CPPIII serves as a biomarker for detecting coproporphyrinogen III, which is rapidly oxidized to CPPIII upon exposure to air during cell lysis and extraction (*Wang et al., 2008*).

## Fractionation of parasite-infected erythrocytes

To assess whether parasites could synthesize heme from ALA in the absence of host enzymes in the erythrocyte cytoplasm, parasite-infected red blood cells were lysed with 0.05% saponin (0.2 µM filtered), spun briefly (3 min, 850×*g*) to pellet, washed in PBS, and resupended in 12 ml of RPMI/albumax growth media supplemented with 200 µM 5-[$^{13}$C$_4$]-ALA. Parasites were incubated overnight at 37°C, harvested by centrifugation, and extracted and analyzed by tandem mass spectrometry as described above.

## Preparation of lysates from uninfected erythrocytes

To assess the heme biosynthesis capacity of the erythrocyte cytoplasm, 500 µl packed red blood cells (described previously [*Sigala et al., 2012*]) were lysed in 20 ml of 0.04% saponin/PBS (0.2 µM filtered) and centrifuged at 25,000×*g* for 60 min to pellet unlysed cells and any organelles. The superficial 15 ml of the lysate supernatant was removed and 0.2 µM filtered, supplemented with 200 µM 5-[$^{13}$C$_4$]-ALA in the absence or presence of 50 µM SA, incubated overnight at 37°C, and analyzed by tandem mass spectrometry as described above.

## Chemical block of parasite nutrient uptake pathways

The 3D7 parasite line expressing the endogenous HSP101 from its genomic locus and bearing a C-terminal *Escherichia coli* DHFR degradation domain (DDD) fusion tag was previously published (*Beck et al., 2014*). To test whether ALA uptake by parasite-infected erythrocytes depends on parasite-established nutrient-acquisition pathways in the host cell membrane, we split a synchronous culture of HSP101-DDD into two populations of late schizonts (purified over a magnetic column) and washed out TMP from one of the two cultures. Both cultures (±TMP) were permitted to lyse and reinvade fresh erythrocytes overnight. The following morning, parasites were placed in 200 µM ALA as early rings, incubated for 8 hr at 37°C, and imaged by live parasite microscopy as described above. Alternatively, asynchronous wild-type 3D7 parasites were incubated in the absence or presence of 100 µM furosemide for 1 hr to block nutrient-acquisition pathways, followed by the addition of 500 µM ALA and further incubation for 8 hr. Parasites were then imaged by live parasite microscopy as described above.

## Antibodies and live cell stains

The following antibodies were used for IFM and WB analysis at the indicated dilutions: goat anti-GFP (Abcam 5450) (IFM 1:500, WB 1:1000), rabbit anti-ACP (*Waller et al., 2000*; *Ponpuak et al., 2007*) (IFM: 1:500), Alexa Fluor 488-conjugated chicken anti-goat (IFM: 1:500), Alexa Fluor 555-conjugated donkey anti-rabbit (IFM: 1:500), donkey anti-goat-IRDye 800 (Licor Biosciences Geneva Lincoln, NE, USA) (WB: 1:10,000). MitoTracker Red was used at 50 nM final concentration and was added to parasite cultures 30 min prior to harvest and analysis.

## Absorbance and luminescence spectroscopy

Fluorescence excitation and emission spectra of pure PPIX in aqueous solution (excitation 400 nm, emission 620 nm) and of clarified lysates of *E. coli* bacteria expressing cobA (excitation 360 nm, emission 600 nm) and chemiluminescence spectra of luminol were obtained on a Cary Eclipse Fluorescence Spectrophotometer (Varian, Inc.) in either fluorescence or luminescence mode using 10-nm slit widths and a PMT detector voltage setting of 600. Luminol solutions contained 25 mM luminol in 100 mM NaOH (aq) to which 0.1% (wt/vol) ammonium persulfate (aq) was added to stimulate light emission.

## Imaging hemozoin dynamics in parasite digestive vacuole

Time-lapse bright field images of parasites were acquired with 1 s delays, composed into video files using Adobe Photoshop, played back with 0.1 s delays between frames, and looped for 15 sec. Bright field images were acquired before and after 4 s image acquisitions with the Zeiss filter sets 38 (excitation 450–490 nm) and 43 HE (excitation 537–562 nm).

## Homology models

Homology models of PBGD and CPO were generated using the SWISS–MODEL interface on the Expasy website (http://swissmodel.expasy.org). The template structures for modeling were human PBGD (PBD: 3EQ1) and human CPO (PDB: 2AEX).

# Acknowledgements

We thank W Beatty for assistance with electron microscopy, A Oksman for assistance with gametocyte culturing, A Vaidya and H Ke for providing ΔFC parasites, and C Roessner for the pISA417 plasmid encoding the cobA gene. We thank J Phillips, D Winge, T Hasan, B Striepen, A Vaidya, I Hamza, and members of the Goldberg lab for helpful discussions. This work was supported by National Institutes of Health grants R21 AI110712-01 (to DEG) and R01 DK099534 (to JPH) and by a Burroughs Wellcome Fund Career Award for Medical Scientists (to JPH) and Career Award at the Scientific Interface (to PAS).

# Additional information

### Competing interests

PAS: Is a co-inventor on a provisional patent application entitled 'Combination Artemisinin and Chemiluminescent Photodynamic Therapy and Uses Therefor'. DEG: Is a co-inventor on a provisional patent application entitled 'Combination Artemisinin and Chemiluminescent Photodynamic Therapy and Uses Therefor'. The other authors declare that no competing interests exist.

### Funding

| Funder | Grant reference | Author |
| --- | --- | --- |
| Burroughs Wellcome Fund | Career Award at the Scientific Interface | Paul A Sigala |
| Burroughs Wellcome Fund | Career Award for Medical Scientists | Jeffrey P Henderson |
| National Institutes of Health (NIH) | R21 AI110712-01 | Daniel E Goldberg |
| National Institutes of Health (NIH) | R01 DK099534 | Jeffrey P Henderson |

The funders had no role in study design, data collection and interpretation, or the decision to submit the work for publication.

### Author contributions

PAS, Conception and design, Acquisition of data, Analysis and interpretation of data, Drafting or revising the article; JRC, Acquisition of data, Analysis and interpretation of data; JPH, Conception and design, Analysis and interpretation of data; DEG, Conception and design, Analysis and interpretation of data, Drafting or revising the article

## Author ORCIDs

Paul A Sigala, http://orcid.org/0000-0002-3464-3042

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
