## [Decision Letter]

Thank you for submitting your work entitled “Deconvoluting Heme Biosynthesis to Target Blood-Stage Malaria Parasites” for peer review at *eLife*. Your submission has been favorably evaluated by Detlef Weigel (Senior editor) and three reviewers, one of whom is a member of our Board of Reviewing Editors. The following individuals responsible for the peer review of your submission have agreed to reveal their identity: Jon Clardy (Reviewing editor); Emily Derbyshire (peer reviewer). A further reviewer remains anonymous.

The reviewers have discussed the reviews with one another and the Reviewing editor has drafted this decision to help you prepare a revised submission.

Summary:

The authors used 5-aminolevulinic acid (ALA), the product of the usual regulatory step in heme biosynthesis, to promote heme biosynthesis in the blood stage parasite. Using a variety of techniques they were able to show that while blood stage parasites do not usually synthesize heme, they can do so upon ALA stimulation. They also showed that host enzymes in the erythrocyte cytoplasm could complement genetic knockouts in the parasite's pathway. With ALA stimulation, a later step in heme biosynthesis become rate limiting, and the build-up of the redox active substrate for this step can inhibit parasite growth. The authors then devised a three-compoent mixture of ALA, luminol, and dihydroartemisinin that potently inhibited parasite growth when used in photodynamic therapy.

Essential revisions:

The reviewers were uniformly enthusiastic about the depth and completeness of the study on heme biosynthesis in the parasite, which utilized a number of techniques to provide a very complete description of the process. There were some concerns about its applied significance as the article points out that heme biosynthesis in the blood stage is normally a non-event, and the biosynthesis studied was prompted by providing a committed substrate, ALA. Please comment on the basic research advances and their relevance to normal malarial infections. In your response, it would be useful to also clarify how your results might, or might not, have been anticipated from earlier work.

The reviewers also appreciated the cleverness behind the development of the potent three-component treatment for photodynamic therapy of malaria with ALA, luminol, and dihydroartemisinin. But there were also concerns about the potential utility of this approach in a typical malaria setting, could you clarify how you think this would work. Toning down the immediate applicability of your findings might be appropriate. In your response to this it would be useful to see a dose response curve for ALA as the concentrations were quite high by usual therapeutic standards.

*Reviewer #1*:

I need to begin by noting that my recent involvement with malaria research focused the liver stage, and that I am not very familiar with the extensive previous research on hemoglobin usage/synthesis in *Plasmodium*. As a result, my review may be off base.

This paper notes that while the parasite genome contains a complete heme biosynthetic pathway that appears to be essential for mosquito and liver stage growth, blood stage parasites with impaired heme biosynthesis show no growth defects because they can scavenge enough heme from the red blood cells they occupy. This manuscript focuses on the effects of stimulating heme biosynthesis in blood stage parasites through the addition of the key starting material 5-aminolevulinic acid (ALA). Formation of ALA is the rate-limiting step in heme biosynthesis, so added ALA bypasses normal heme biosynthetic regulation. This article focuses on two points: heme biosynthesis in infected red blood cells and a potential new therapeutic approach to treating malaria infections.

Heme biosynthesis: ALA-stimulated biosynthesis is observed even in erythrocytes infected with parasites in which key heme biosynthetic enzymes have been knocked out. The authors resolve this seeming paradox by establishing that host enzymes in the infected erythrocyte's cytoplasm can compensate the genetic knockouts. I believe that this finding is firmly established, but not very surprising.

Malaria therapy: Prior work had shown that ALA-stimulated biosynthesis leads to a new rate-limiting step and the accumulation of the starting material (PPIX) for this step. PPIX is fluorescent and this fluorescence has previously been used for following heme biosynthesis. The authors establish that parasites grown in 200 µM ALA fluoresce brightly. Prior work had also established that PPIX produced reactive oxygen species (ROS) when irradiated, and the cytotoxic effects of ROS generation has been exploited as a photodynamic therapy for cancer cells. In accord with this precedent, irradiation of parasite cultures treated with ALA displayed a significant growth defect. The authors explore the mechanism of growth inhibition in considerable detail and establish that host enzymes are largely responsible for PPIX formation and that parasites have an ALA uptake mechanism. They then go on to develop a photodynamic therapy for malaria using earlier findings about luminol enhancement from cancer chemotherapy and introducing artemisinin with its obvious relevance to malaria therapy. A three-component mixture of ALA, luminol, and dihydroartemisinin was potent in in vitro assays.

Overall, I'm impressed with the thoroughness of the work, but I don't feel that any of the findings reach the level of significance usually found in *eLife* publications. In particular, I find the claims that this could lead to a useful therapeutic treatment for malaria to be inflated. The assays are in vitro, and the concentrations (100 µM ALA and 750 µM luminol) seem very high. The criteria for malaria therapies, which need to be effective in resource poor settings, include: single dose, low cost (preferably less than $1), efficacy in all stages, and safety for pregnant women and children. Recently important articles on just such therapies have appeared (Nature) as have articles on significant host targets (Science). There is a high bar for innovative and useful malaria therapies. I'm inclined to recommend against publication.

*Reviewer #2*:

The manuscript by Sigala et al. uses a combination of state of the art biochemistry, gene disruption, metabolic LC/MS studies with 13C and cell imaging to dissect the heme biosynthesis pathway in *Plasmodium* erythrocytic stages and importantly to determine the contribution of parasite versus host red cell enzymes in the parasite biology. The authors show conclusively that the parasite heme biosynthesis is not essential through evaluation of several genetic knockouts in the pathway (e.g. porphbilinogen deaminase and cytosolic corproporphyrinogen III oxidase). They demonstrate that heme biosynthesis is stimulated by addition of exogenouse 5-aminolevulinic acids (ALA) and they show through fractionation and 13C labeling that host red cell enzymes and not parasite enzymes are responsible for metabolism of exogenous ALA to generate protoprphyrin IX (PPIX). They also show that parasite remodeling of the host cell is necessary to permeabilize the red cells to ALA uptake, thus there is also an important contribution of parasite expressed proteins to the ability of ALA to be utilized by the infected red cell. The identification of the role of red cell enzymes explains the paradox that *Plasmodium* ferrochelatase is essential while the upstream heme biosynthetic enzymes from the parasite are not. Finally they show that combinations of ALA, luminol and an activator such as artemesinin can be used to kill the parasites in vitro in the absence of light stimulation. (PPIX generates cytotoxic reactive oxygen species when photo-illuminated). Photodynamic therapy has been explored as a mechanism to kill cancer cells treated with ALA. As conventional photodynamic therapy would require a light source and not be practical for treatment of malaria, the authors used a combination of luminol activated by an oxygen radical generator (e.g. Artemisinin) to demonstrate that parasites are killed by these in combination with ALA.

This study very nicely solves the conundrum of heme biosynthesis in blood stage *Plasmodium*, demonstrating conclusively that it is not essential, that the parasites use instead salvaged heme, and demonstrating conclusively that stimulation of PPIX production in the presence of ALA is mediated by host cell enzymes. The described work is highly innovative and creative, and the logic of the approach is impressive, well-reasoned and well laid out in the manuscript. The work is very carefully done and uses a strong combination of approaches that lead to clear conclusive results. The possibility that photodynamic therapy might be used to treat malaria is also intriguing and the use of artemisinin as the oxygen radical source was very clever and novel. The study provides a mechanism for potentially rescuing the value of artemisinin even in “resistant” parasites. This paper clearly advances in a very significant way our understanding of heme biosynthesis in the malaria parasite and it provides a provocative potential new method for parasite treatment. The work is provocative and of excellent quality and I fully support its publication.

I have a few comments/questions that should be considered in the revision.

1) Data described in the third paragraph of subsection “Development of chemiluminescence-based photodynamic therapy for treatment of blood-stage malaria”. Were parasites in this study protected from light to ensure that there was no exogenous light induced activation of the killing mechanism?

2) I think more could be said in the Discussion about the practicality of using photodynamic therapy in vivo, including a better sense of if this method has worked in vivo for cancer. In the fifth paragraph of the Discussion they state that ALA and luminol have good toxicity profiles, but what about the PK and oral bioavailbility profiles, is there any data to address if these profiles would be suitable for malaria therapy?

3) In the fourth paragraph of the Discussion, it should be clarified whether or not the authors think 0.2 uM exogenous ALA would be sufficient to induce light induced killing? Particularly given the use of 100 - 200 uM in their studies. Was any dose titration done to determine the lowest level of ALA that could be used for light induced killing? How was the dose of ALA chosen for the studies?

*Reviewer #3*:

In the manuscript entitled “Deconvoluting Heme Biosynthesis to Target Blood Stage Malaria Parasites” Sigala et al. use biochemical, genetic and chemical methods to unravel the role of heme biosynthesis in blood stage malaria. Their work demonstrates that heme biosynthesis de novo is dispensable for parasite viability, but can occur after addition of the heme precursor ALA using enzymes in both the parasite and host erythrocyte. The latter being particularly interesting since mature red blood cells lose part of their heme biosynthesis pathway when they expel their mitochondria. The manuscript determines that parasites enable the uptake of ALA into the erythrocyte with established new permeability pathways and demonstrate that the combination of luminol and ALA is lethal. This interesting work will greatly contribute to the current body of literature on *Plasmodium* heme biosynthesis and presents a novel malaria treatment strategy.

The manuscript clearly explains how their proposed luminol/ALA combination therapy would target a host process and therefore may not lead to parasite drug resistance. Since this process relies on host processes it also seems likely that such a strategy would be effective against multiple human-infective parasite species. While the authors do not test this it might be worth mentioning in the Discussion.

*Minor comments*:

1) It might be helpful to readers to indicate on the heme biosynthesis schematic (Figure 1) that SA inhibits ALAD.

2) In the third paragraph of the Discussion it is mentioned that human serum contains 100 - 200 nM ALA and there is data that suggests that this low level is enough to stimulate heme biosynthesis. Based on Figure 7 this amount does not seem like it is enough to be toxic with luminol addition. For many experiments in this work 200 uM ALA was added to the cultures and is sufficient to induce toxicity. It would be interesting to know the lower limit of ALA addition that can induce toxicity if it was determined.

---

## [Author Response]

*The reviewers were uniformly enthusiastic about the depth and completeness of the study on heme biosynthesis in the parasite, which utilized a number of techniques to provide a very complete description of the process. There were some concerns about its applied significance as the article points out that heme biosynthesis in the blood stage is normally a non-event, and the biosynthesis studied was prompted by providing a committed substrate, ALA. Please comment on the basic research advances and their relevance to normal malarial infections. In your response, it would be useful to also clarify how your results might, or might not, have been anticipated from earlier work*.

Genome sequencing has been enormously helpful in identifying the metabolic pathways potentially used by *Plasmodium* parasites to survive and proliferate within human erythrocytes and which might serve as new therapeutic targets. Nevertheless, metabolic maps resulting from gene annotation are only a model, and a full understanding of parasite metabolism and the therapeutic potential of individual pathways requires direct genetic and biochemical dissection that goes well beyond knocking out individual genes or testing the effects of inhibitors assumed to be specific based on studies in other organisms.

Our work clarifies that the parasite heme biosynthesis pathway, despite expression of all pathway enzymes, is non-functional during blood-stage infection and that porphyrin biosynthesis activity by parasite-infected erythrocytes arises almost entirely from host enzyme activity in the red cell cytoplasm. This surprising result could not have been anticipated from prior works (e.g., Sigala 2014, van Dooren 2012, Surolia 1992, Ke 2014, Nagaraj 2013), all of which have assumed activity by the parasite heme biosynthesis pathway in blood stages.

Although the remnant host enzymes are normally inactive in uninfected erythrocytes, due to lack of the ALA substrate resulting from the absence of mitochondrial ALAS as well as erythrocyte impermeability to low-level serum ALA, our data suggests that this host-derived pathway can be active in the context of a parasite-infected erythrocyte, whose enhanced permeability enables uptake of ∼0.2 µM serum ALA. This low-level ALA is likely sufficient to stimulate modest heme biosynthesis in the parasite mitochondrion, where the parasite ferrochelatase can insert iron into the PPIX synthesized from host enzyme activity (see Figure 4). Our work thus clarifies the fundamental mechanism of heme biosynthesis in parasite-infected erythrocytes, which reflects the concerted activity of ALA-stimulated host enzymes plus the parasite ferrochelatase.

Most drug discovery studies focus nearly exclusively on the suitability of a metabolic pathway for therapeutic inhibition. If a pathway is not essential then it is usually not pursued as a drug target. Though heme biosynthesis is not an essential pathway for blood-stage parasites (Ke 2014, Nagaraj 2013), our manuscript shows that this pathway is nonetheless targetable because stimulation of pathway activity can be exploited to kill parasites. The success of our strategy could not have been anticipated from earlier work, as recent work (Ke 2014, Nagaraj 2013) concluded that this pathway was not a druggable target.

In summary, our detailed study combining both genetic and metabolic dissection has both clarified basic biochemical activity and mechanism of a fundamental biosynthetic pathway and suggested the potential for a novel form of antimalarial therapy. We have modified the Discussion to clarify these points.

*The reviewers also appreciated the cleverness behind the development of the potent three-component treatment for photodynamic therapy of malaria with ALA, luminol, and dihydroartemisinin. But there were also concerns about the potential utility of this approach in a typical malaria setting, could you clarify how you think this would work. Toning down the immediate applicability of your findings might be appropriate*.

Dihydroartemisinin (DHA) is already in clinical use for treatment of blood-stage malaria. We envision a combination therapy in which appropriate ALA and luminol doses are delivered orally with DHA. As for any new therapy, extensive additional studies in animals and humans will be required to optimize treatment regimens for our proposed therapy and to confirm efficacy and safety in vivo. We expound on these points below in responses to specific Reviewer points. We have modified the Discussion to clarify that additional in vivo studies will be required to further test and optimize our proposed therapy for clinical use.

*In your response to this it would be useful to see a dose response curve for ALA as the concentrations were quite high by usual therapeutic standards*.

As noted below in our response to Reviewer #3 (point 2), we have now inserted an ALA-dose response curve for parasite photosensitivity as Figure 1—figure supplement 3.

With respect to “therapeutic standards”, we note that the 10-100 µM ALA used herein (e.g., Figure 7 and Figure 7 and Figure 1—figure supplement 3) is well within ALA concentrations that have been tested systemically in vivo in humans for clinical photodynamic therapy and that appear to be well tolerated. Indeed, plasma ALA concentrations ∼10 µM can be obtained in humans by a single oral dose of 100 mg ALA, which is well below the 40-50 mg/kg doses of ALA that have been used clinically with minimal side effects. These pharmacokinetic properties are explored in J Pharmacol Exp Ther 2002: 507-512, Photochem Photobiol 2001: 339-345, and references therein, and we now cite these references in the Discussion section.

Reviewer #1:

*[…] Overall, I'm impressed with the thoroughness of the work, but I don't feel that any of the findings reach the level of significance usually found in* eLife *publications. In particular, I find the claims that this could lead to a useful therapeutic treatment for malaria to be inflated. The assays are in vitro, and the concentrations (100 µM ALA and 750 µM luminol) seem very high. The criteria for malaria therapies, which need to be effective in resource poor settings, include: single dose, low cost (preferably less than $1), efficacy in all stages, and safety for pregnant women and children. Recently important articles on just such therapies have appeared (Nature) as have articles on significant host targets (Science). There is a high bar for innovative and useful malaria therapies. I'm inclined to recommend against publication*.

Please see our responses above (Essential revisions) and below to Reviewer #2 (point 2) and reviewer #3 (point 2).

Reviewer #2:

*1) Data described in the third paragraph of subsection “Development of chemiluminescence-based photodynamic therapy for treatment of blood-stage malaria”*. *Were parasites in this study protected from light to ensure that there was no exogenous light induced activation of the killing mechanism?*

We minimized the exposure of parasite cultures to ambient light by changing media within darkened TC hoods and covering parasite culture dishes during brief (5-10 sec.) transits to and from incubators. As shown in Figure 7 and Figure 7, this low level of light exposure was insufficient to activate phototoxicity in the ALA-only control culture. We have clarified this point in the Methods.

*2) I think more could be said in the Discussion about the practicality of using photodynamic therapy in vivo, including a better sense of if this method has worked in vivo for cancer. In the fifth paragraph of the Discussion they state that ALA and luminol have good toxicity profiles, but what about the PK and oral bioavailbility profiles, is there any data to address if these profiles would be suitable for malaria therapy*?

There is an extensive literature regarding the use and efficacy of photodynamic therapy (PDT) for treatment of cancer, including use of ALA as a photosensitizer. We refer the reviewer and interested readers to Kennedy 1990 and Celli 2010 and Photochem Photobiol Sci 2007:1234-1245, which discuss PDT and its application for cancer therapy in depth, and to references therein. Bioavailability and PK data for systemic use of ALA and luminol are available in the literature (e.g., J Pharmacol Exp Ther 2002: 507-512, Molecules 2011: 4140-4164, Photochem Photobiol 2001: 339-345, reference 49, and references therein) including data to suggest that ALA doses sufficient to attain serum concentrations in the range explored in vitro herein persist for several hours in the blood-stream and show little associated toxicity (see also responses to Essential Revisions above).

As for any new therapy suggested by in vitro studies, extensive additional studies in animals and humans will be required to optimize treatment regimens for our proposed therapy and to confirm efficacy and safety in vivo. We have plans to carry out these experiments, beginning with in vivo studies with the mouse malaria parasite, *P. berghei.* The results of these studies will be the subject of future manuscripts, as they extend well beyond the scope of the present study. Our goals in this manuscript were first and foremost to clarify the basic biochemical mechanisms and properties of heme biosynthesis in *Plasmodium*-infected erythrocytes and secondarily to explore whether ALA-stimulated porphyrin biosynthesis could be harnessed to target blood-stage malaria. Our results herein using artemisinin and chemiluminescence-based photodynamic therapy show outstanding promise in vitro, and we are moving forward to test this strategy in vivo.

We have inserted the additional references noted above regarding the use of PDT for cancer treatment and the bioavailability and PK features of ALA and luminol. We have also modified the Discussion to clarify that additional in vivo studies will be required to further test and optimize our proposed therapy for clinical use.

*3) In the fourth paragraph of the Discussion, it should be clarified whether or not the authors think 0.2 uM exogenous ALA would be sufficient to induce light induced killing? Particularly given the use of 100 - 200 uM in their studies. Was any dose titration done to determine the lowest level of ALA that could be used for light induced killing? How was the dose of ALA chosen for the studies*?

Please see our response below to Reviewer #3, point 2.

Reviewer #3:

*[…] The manuscript clearly explains how their proposed luminol/ALA combination therapy would target a host process and therefore may not lead to parasite drug resistance. Since this process relies on host processes it also seems likely that such a strategy would be effective against multiple human-infective parasite species. While the authors do not test this it might be worth mentioning in the Discussion*.

We agree with the reviewer that our proposed therapy may be active against other pathogens that infect human erythrocytes, such as *Babesia*. Because ALA uptake requires enhanced erythrocyte permeability upon parasite infection, it remains a future challenge to test whether other intraerythrocytic parasites like *Babesia* also alter host cell permeability in a manner that enables ALA uptake. We have followed the reviewer’s suggestion and have modified the Discussion section to propose that chemiluminescence-based photodynamic therapy may be active against other parasites that infect erythrocytes.

Minor comments:

*1) It might be helpful to readers to indicate on the heme biosynthesis schematic (*Figure 1*) that SA inhibits ALAD*.

We thank the reviewer for this suggestion. We have modified both Figure 1 and Figure 4 to indicate that succinylacetone inhibits ALAD.

*2) In the third paragraph of the Discussion it is mentioned that human serum contains 100 - 200 nM ALA and there is data that suggests that this low level is enough to stimulate heme biosynthesis. Based on*
Figure 7
*this amount does not seem like it is enough to be toxic with luminol addition. For many experiments in this work 200 uM ALA was added to the cultures and is sufficient to induce toxicity. It would be interesting to know the lower limit of ALA addition that can induce toxicity if it was determined*.

We looked at the ALA dose-dependence of photosensitivity by parasites. Parasite death and culture growth inhibition in the presence of light was detectable in ALA concentrations as low as 5-10 µM, although full or nearly full growth inhibition required ALA concentrations greater than 50 µM. We have added this data as Figure 1—figure supplement 3.